# Information Processing with Stability Point Modeling in Cohen–Grossberg Neural Networks

## Ekaterina Gospodinova *,† and Ivan Torlakov †

Faculty of Engineering and Pedagogy of Sliven, Technical University of Sofia, 1756 Sofia, Bulgaria; itorlakov@tu-sofia.bg

* Correspondence: ekaterina_gospodinova@tu-sofia.bg
† These authors contributed equally to this work.

**Abstract:** The aim of this article is to develop efficient methods of expressing multilevel structured information from various modalities (images, speech, and text) in order to naturally duplicate the structure as it occurs in the human brain. A number of theoretical and practical issues, including the creation of a mathematical model with a stability point, an algorithm, and software implementation for the processing of offline information; the representation of neural networks; and long-term synchronization of the various modalities, must be resolved in order to achieve the goal. An artificial neural network (ANN) of the Cohen–Grossberg type was used to accomplish the objectives. The research techniques reported herein are based on the theory of pattern recognition, as well as speech, text, and image processing algorithms.

**Keywords:** neural network; mathematical modeling; Cohen–Grossberg; structured information; algorithm

## 1. Introduction

Some problems in the field of pattern recognition have been successfully solved. Commercial systems for speech recognition, image recognition, and automatic text analysis are known. The degree of success in solving these problems depends on the degree of formalized description of the subject area [1]. Face image recognition is solved in isolation. The problem of identifying grammar and syntax errors is a more complex task. Recognition of images and scenes, dictation of texts from a microphone, and automatic classification of text are unsolved tasks. Existing systems only demonstrate their level of complexity. The difficulties that arise in solving these problems are in the synchronization of the analyzed information. This leads to the formation of large, pure hypotheses. In the case of the processing and synchronization of a large amount of information, their verification becomes a non-trivial task and is also unsolved within the framework of the applied methods. Currently, the complexity of the methods for representing semantic and mathematical information with both metalinguistic and figurative means practically does not allow for their effective use to solve problems. Within the scientific direction of artificial intelligence, numerous attempts have been made and are being made to use semantic and pragmatic information, mainly to solve the problem of human–machine communication in natural language. The works of Alkon, Alwang, and Bengio [2–4] are widely known. Their success is due to the fact that the semantic picture is replaced by the rigid structure of the relational database, from which natural language interpretations are made and attempts to interpret statements in terms of concepts are made. However, the great ambiguity of these interpretations arises from the inaccuracy of language models. It is not possible to automatically form a model based on texts alone. Less well-known are the ways in which semantic information is used. Image recognition Quasi-Zo [4,5] has been used as a

world model to analyze scenes in which individual objects are represented by generalized geometric shapes such as balls and cylinders. With the help of this model, objects in the scene are represented, segmented, and identified and further described in metalinguistic terms, as well as the relations between them and their dynamics. Furthermore, all these steps are processed separately.

The development of methods for the representation of information at the semantic and pragmatic levels (equally convenient for both linguistic and image recognition tasks) is a key point in improving both the quality and functionality of these systems and in the transition to the next stage of development of intelligent systems (IS), i.e., the stage of creating integrated multimodal systems for information processing and storage. The existence of these tasks makes us look for new approaches to the methods of presenting and processing information from different modalities—verbal, visual, and supermodal (semantic and pragmatic) information—in synchrony [6].

Introducing knowledge into artificial IS is effective not due to modeling of individual intellectual functions but due to modeling of the computing environment in which entire tasks are solved. Intellectual systems are those that perform intellectual functions within the framework of cognitive behavior: perception, learning, formation of thinking patterns (using a pattern to solve current problems), problem solving, prediction, decision making, linguistic behavior, etc. Therefore, IS include natural language processing information systems, word processing systems, and automated systems. The classification of existing systems allows them to be divided into two classes: single-level systems that recognize speech events using one-way or modified Bayesian rules (implemented on a neural network) and synchronous processing systems using empirical linguistic rules [7].

For example, after an acoustic speech signal is fed into the system, it is digitized, cleaned up, normalized by amplitude, and freed from related information. Then, its pieces are compared to the standards for each level that were made during the training stage.

In the case of solving simple recognition problems, commands with a limited vocabulary and single-level statistical approaches are most often used. In order to solve more complicated problems, such as finding keywords in a stream of continuous speech, the structural approach needs to use information from all levels of language, from morphology to syntax, as well as information from outside of languages, such as semantics and pragmatics. The complexity of the task of building speech recognition systems comes from the fact that a lot of information with different internal structures to be processed by different algorithms needs to be put together into a single whole. In addition, the use of practical solutions to the problem of speech recognition encounters a psychological barrier, which consists of the fact that a person expects the same possibilities in communication from speech recognition systems as in communication with a person. Solving the latter task involves recreating—if possible—processing, and presenting the information at one's disposal. This means that in addition to integrating linguistic and extralinguistic sources of knowledge at different levels, it is necessary to integrate information processing subsystems from other modalities—primarily visual. When three problems are solved, effective synchronization and integration of a large amount of disparate information become possible. First, it is necessary to use the same algorithms to process information with different structures. Second, it is desirable to implement these algorithms with the use of specialized (directed precisely to these algorithms) equipment instead of universal processing means. Thirdly, it is necessary to implement an associative means of synchronizing information.

Analysis of existing systems showed that, as in the case of speech recognition, when solving the problem of image recognition, two main approaches are used: geometric and linguistic. Image recognition has its own problems when it comes to sorting through large quantities of information because it requires a considerable amount of math. IS that work well can only be made if they are synchronized and have a high level of resilience. They can be represented in the form of a set of rules or a declarative representation of knowledge when the information is represented in the form of a database. Solving the problem of

integrating information from different modalities would allow us to escape this vicious circle [8–10].

The aim of this work is to identify effective ways of synchronizing multilevel structured information from different modalities (images, speech, and text), which allows for natural reproduction of the structure of information as it occurs in the human brain. Processing optimization methods should allow for modeling of a sustainable process. For this purpose, neural network representation and processing of different modalities can be used. It is also necessary to develop a method and algorithm to train neural networks for robustness and synchronization.

The current paper contains five sections in which we described the used of CGNN training to analyze time sequences to represent speech as textual information, resulting in h stability for dynamic information generation. Then, a description of the algorithm and an example implementation of a stability model in a CGNN are presented.

## 2. Cohen–Grossberg Network Training for Different Modalities

Using an ANN and considering existing solutions showed that they can be broken down into two types: static and dynamic systems. Classical networks with elements that are like neurons can solve the problem of recognizing spatial images and speech characteristics. Dynamic images and speech can also be recognized by using networks with delay elements and dynamic neural networks. In this case, special techniques are used to take into account the way information is organized over time.

An ANN that takes into account dynamic time-based information can be used to analyze temporal sequences in which the representation of both speech and visual/textual information is reduced, such as a Cohen–Grossberg network.

Cohen and Grossberg presented their variation of an ANN in [11] represents self-organization, competitiveness, etc. Grossberg designed a continuous time-based racing network based on the human visual system. His work is characterized by the use of non-linear mathematics to model specific functions. The topics of their papers include specific areas, such as how adversarial networks can provide an improvement in recognizable information in vision, and their work is characterized by a high level of mathematical complexity [12,13].

In order to take the temporal structure of the information into account, a special technique is used. Information is fed with delays due to additional network inputs, and the highest output is expected to be produced. In this case, the network begins to take into account the time context of the input, and dynamic images are automatically formed.

We propose the use of the learning law of the adaptive weights in the Grossberg network, which W. Grossberg calls long-term memory (LTM) because the rows of $W$ represent patterns that have been stored and can be recognized by the network. The stored pattern that is closest to the input produces the highest output in the second layer.

One law of learning for $W^2$ is given by:

$$\frac{dw_{i,j}^2(t)}{dt} = \alpha\left\{-w_{i,j}^2(t) + n_i^2(t)n_j^1(t)\right\} \tag{1}$$

where $\alpha$ is yjr learning rate coefficient, $W$ is the input vector, and $t$ is the time variable. The two-layer equations show a passive decay term in the first term in the bracket on the right and a Hebbian-like learning process in the second term. Combined, these terms result in the deterioration of Hebb's rule.

The first-layer equation normalizes the strength of an input pattern after receiving external inputs. The input vector ($p$) is used to calculate the excitatory and inhibitory inputs. It takes the shape of

$$\varepsilon\frac{dn^1(t)}{dt} = -n^1(t) + \left({}^+b^1 - n^1(t)\right)\left[{}^+W^1\right]p - \left(n^1(t) + {}^-b^1\right)\left[{}^-W^1\right]p \tag{2}$$

where $\varepsilon$ determines the speed of response, $p$ is the input vector, $t$ is the time variable, and $b$ is inhibitory bias.

Equation (2) is an intriguing shunt model with the following input:

$$^{+}W^1 = \begin{bmatrix} 1 & 0 & \dots & 0 \\ 0 & 1 & \dots & 0 \\ \vdots & \vdots & & \vdots \\ 0 & 0 & \dots & 1 \end{bmatrix} \tag{3}$$

The sum of each element in the input vector—aside from the $i$th element—therefore constitutes the inhibitory input to the $i$th neuron.

The on-center/off-surround pattern is created by the two matrices: $^{-}W^1$ and $^{+}W^1$ because the inhibitory input, which shuts the neuron off, comes from locations outside of the input vector, while the excitatory input, which includes the neuron, comes from the $i$th element of the input vector, which is centered at the same point. The input pattern is normalized by this style of binding pattern.

The lower bound of the maneuver pattern is set to zero by setting the inhibitory bias ($^{-}b^1$) to zero for the sake of simplicity. Moreover, it uniformly adjusts all components of the excitation bias ($^{+}b^1$):

$$^{+}b_i^1 = {}^{+}b^1, \quad i = 1, 2, \dots, S^1 \tag{4}$$

As a result, the upper bound for each neuron is equal. Let us examine the first layer's normalizing effect, where the $i$th neuron's response has the following form:

$$\varepsilon \frac{dn_i^1(t)}{dt} = -n_i^1(t) + \left({}^{+}b^1 - n_i^1(t)\right) p_i - n_i^1(t) \sum_{j \neq i} p_j \tag{5}$$

At steady state, $\left(dn_i^1(t)/dt = 0\right)$ gives us:

$$0 = -n_i^1 + \left({}^{+}b^1 - n_i^1\right) p_i - n_i^1 \sum_{j \neq i} p_j \tag{6}$$

If we opt for neuron $n_i^1$ steady-state output, the outcome is:

$$n_i^1 = \frac{{}^{+}b^1 p_i}{1 + \sum\limits_{j=1}^{S^1} p_j}. \tag{7}$$

The relative intensity of the $i$th input is defined as follows:

$$\bar{p}_i = \frac{p_i}{P} \text{ where } P = \sum_{j=1}^{S^1} p_j \tag{8}$$

The steady-state activity of neurons takes the following form:

$$n_i^1 = \left(\frac{{}^{+}b^1 P}{1 + P}\right) \bar{p}_i \tag{9}$$

Hence, regardless of the size of the overall input ($P$), $n_i^1$ is always proportional to the relative intensity ($\bar{p}_i$). Moreover, the neuron's overall activity is modest:

$$\sum_{j=1}^{S^1} n_j^1 = \sum_{j=1}^{S^1} \left(\frac{{}^{+}b^1 P}{1 + P}\right) \bar{p}_j = \left(\frac{{}^{+}b^1 P}{1 + P}\right) \leq {}^{+}b^1 \tag{10}$$

The input vector is normalized to maintain the relative intensities of its separate components while reducing the overall activity to less than $^{+}b$. As a result, rather than encoding the instantaneous variations in the total input activity ($P$), the first layer's outputs ($n_i^1$) encode the relative input intensities ($\bar{p}_i$). This outcome is the result of the shunt model's nonlinear gain control and the on-center/off-surround coupling of the inputs.

The consistency of the information that has been processed and the dynamic properties of the visual system are explained in the first layer of the Grossberg network. The network responds to relative, not absolute, picture intensities.

The continuous-time period layer, the second layer of the Grossberg network, serves a number of purposes. The entire activity in the layer is first normalized, just like the first layer. Second, the detected information enhances its model, making it more likely that the neuron with the greatest input also produces the strongest response. Lastly, it stores the amplified model, acting as short-term memory (STM).

The presence of feedback in the second layer is the primary distinction between the two layers. It enables the network to retain a pattern even when the input is no longer present. The band also engages in competition, which amplifies the information that may be recognized in the pattern.

The equation for the second layer takes the following form:

$$\varepsilon \frac{dn^2(t)}{dt} = -n^2(t) + \left(^{+}b^2 - n^2(t)\right)\left\{\left[^{+}W^2\right]f^2\left(n^2(t)\right) + W^2a^1\right\}$$
$$- \left(n^2(t) + {}^{-}b^2\right)\left[{}^{-}W^2\right]f^2\left(n^2(t)\right) \tag{11}$$

This is a shunt model with an excitation input of $\left\{\left[^{+}W^2\right]f^2\left(n^2(t)\right) + W^2a^1\right\}$, while on-center feedback is expressed as $^{+}W^2 = {}^{+}W^1$, and adaptive weights similar to those in a Kohonen network make up $W^2$. Following training, the rows of $W^2$ indicate prototype models. The off-surround feedback provided by $\left[{}^{-}W^2\right]f^2\left(n^2(t)\right)$ is the inhibitory input to the shunting model. $^{-}W^2 = {}^{-}W^1$ provides this feedback.

The following example of a network with two neurons can be taken into consideration to demonstrate the impact of the second layer of a Grossberg network:

$$\varepsilon = 0.1 \quad {}^{+}b^2 = \begin{bmatrix} 1 \\ 1 \end{bmatrix} \quad W^2 = \begin{bmatrix} (_1w^2)^T \\ (_2w^2)^T \end{bmatrix} = \begin{bmatrix} 0.9 & 0.45 \\ 0.45 & 0.9 \end{bmatrix} \tag{12}$$

and

$$f^2(n) = \frac{10(n)^2}{1 + (n)^2} \tag{13}$$

The layer equations are:

$$(0.1)\frac{dn_1^2(t)}{dt} = -n_1^2(t) + \left(1 - n_1^2(t)\right)\left\{f^2\left(n_1^2(t)\right) + \left(_1w^2\right)^T a^1\right\}$$
$$- n_1^2(t)f^2\left(n_2^2(t)\right) \tag{14}$$

$$(0.1)\frac{dn_2^2(t)}{dt} = -n_2^2(t) + \left(1 - n_2^2(t)\right)\left\{f^2\left(n_2^2(t)\right) + \left(_2w^2\right)^T a^1\right\}$$
$$- n_2^2(t)f^2\left(n_1^2(t)\right) \tag{15}$$

The prototype models (rows of the weight matrix ($W^2$)) and the output of the first layer serve as the internal multipliers for the second layer (normalized input model). The prototype model that is most similar to the input model has the highest internal multiplier. The second layer then engages in competition among neurons, which has the effect of supporting large outputs while attenuating small outputs, thereby tending to improve the

output pattern. Competition in a Grossberg network preserves large values while reducing small values, yet it need not necessarily reduce all small values to zero. The activation function controls how much recognizable information is amplified [11].

Two key characteristics are mentioned. First, some information augmentation occurs before the input is eliminated. According to the second layer's inputs:

$$(_1w^2)^T a^1 = \begin{bmatrix} 0.9 & 0.45 \end{bmatrix} \begin{bmatrix} 0.2 \\ 0.8 \end{bmatrix} = 0.54 \tag{16}$$

$$(_2w^2)^T a^1 = \begin{bmatrix} 0.45 & 0.9 \end{bmatrix} \begin{bmatrix} 0.2 \\ 0.8 \end{bmatrix} = 0.81 \tag{17}$$

As a result, the input to the second neuron is 1.5 times that of the first neuron. However, after a quarter of a second, the second neuron's output surpasses that of the first neuron by a factor of 6.34.

The network subsequently develops and saves the pattern once the input is set to zero, which is the second distinguishing feature of the response. Even after the input is stopped, the output continues. Grossberg [11] refers to this tendency as reverberation. The network can store the pattern and the on-center versus off-surround pattern of the connections, which are determined by $^+W^2$ and $^-W^2$, thanks to nonlinear feedback. This leads to an improvement.

It is taken into consideration that both levels of the Grossberg network use an on-center/off-surround structure [14]. Other connection patterns are available for usage in various applications. The directed receptive field has been suggested as a structure to implement this technique [15]. The "on" (excitatory) connections for this structure originate from one side of the field, whereas the "off" (inhibitory) connections originate from the other side of the field.

When $n_i^2(t)$ is not active, it is feasible to disable learning in specific circumstances. The equation in this case has the following training form:

$$\frac{dw_{i,j}^2(t)}{dt} = \alpha n_i^2(t) \left\{ -w_{i,j}^2 + n_j^1(t) \right\}, \tag{18}$$

which is expressed in the form of a vector as

$$\frac{d\left[ _iw^2(t) \right]}{dt} = \alpha n_i^2(t) \left\{ -\left[ _iw^2(t) \right] + n^1(t) \right\} \tag{19}$$

where $\left[ _iw^2(t) \right]$. The elements of the *i*th row of $W^2$ make up the vector.

Learning is only possible when the terms on the right-hand side of Equation (1) are multiplied by the integer $n_i^2(t)$. This is an ongoing application of the principle of learning from the beginning. The topology and structure of the data being converted are preserved because of the learning law of adaptive weights. Similar pieces are converted along the same trajectory, whereas distinct fragments are converted along various paths. In this scenario, the network starts to consider the temporal context of the input. Then, it is possible to automatically create dynamic picture standards [12,13].

## 3. H-Stability Results

The recurrent ANN performed well when used to handle temporal information; however, the act of manually transforming the structure into recognizable data is where the issue lies. A neural network that performs multilevel information processing must make use of the robustness concept with regard to manifolds and robustness criteria in order to successfully resolve this issue [16]. ANNs are useful for the study of temporal sequences in which the presentation of speech, visual, and textual information is condensed. In these networks, impulse events are realized at a given moment and can be derived as a result of the current H-stability result. The major findings concerning the equilibrium state of the

(20) model's $h$ stability are taken into consideration. The authored lemma and authored theorem are found in [16–18].

Equation (20) was taken from [17], which presents the theoretical model mentioned in Theorem 1 from the same paper.

$$
\begin{cases}
\dot{x}_i(t) = -a_i(x_i(t))\left[b_i(x_i(t)) - \sum_{j=1}^{m} c_{ij}f_j(y_j(t))\right. \\
\qquad\qquad \left. - \sum_{j=1}^{m} d_{ij}g_j(y_j(t - \sigma_j(t))) - I_i\right], t \neq \tau_k(x(t), y(t)), \\
\dot{y}_j(t) = -\hat{a}_j(y_j(t))\left[\hat{b}_i(x_i(t)) \sum_{j=1}^{n} p_{ij}\hat{f}_j(x_i(t))\right. \\
\qquad\qquad \left. - \sum_{i=1}^{n} q_{ij}\hat{g}_j(x_i(t - \hat{\sigma}_j(t))) - J_j\right], t \neq \tau_k(x(t), y(t)), \\
(x_i(t^+), \ y_j(t^+))^T = (x_i(t) + P_{ik}(x_i(t), y_j(t) + Q_{jk}(y_j)))^T, \\
\qquad\qquad t \neq \tau_k(x(t), y(t)),
\end{cases}
\tag{20}
$$

**Theorem 1.** *Let us assume that:*

1.  *There exists a positive number ($\mu$), and*

$$
\begin{aligned}
&min_{1 \leq i \leq n}\left(\underline{a_i}B_i - \overline{a_j}\sum_{j=1}^{m}|p_{ij}||\hat{L}_i|\right) + min_{1 \leq j \leq m}\left(\hat{a}_j\hat{B}_j - \hat{\overline{a}}_j\sum_{i=1}^{n}|c_{ij}||L_j|\right) \\
&- \left(max_{1 \leq j \leq m}\overline{a_j}\sum_{i=1}^{n}|d_{ij}|M_j + max_{1 \leq i \leq n}\hat{\overline{a}}_i\sum_{j=1}^{m}|q_{ij}||\hat{M}_i|\right) \geq \mu;
\end{aligned}
\tag{21}
$$

2.  *The functions $P_{ik}$ and $Q_{jk}$ are such that*

$$
P_{ik}(x_i(t_k)) = -\gamma_{ik}(x_i(t_k) - x_i^*), Q_{jk}(y_j(t_k)) = -\mu_{jk}(y_j(t_k) - y_j^*),
$$

*where $0 < \gamma_{ik} < 2, 0 < \mu_{jk} < 2, i = 1, 2, \ldots, n, j = 1, 2, \ldots, m, k = 1, 2, \ldots;$*

3.  *Such a function exists ($h(t, z)$), where the following inequalities hold [19–21]:*

$$
\|h(t, z)\| \leq \|z\| < \Lambda(H)\|h(t, z)\|, (t, z) \in [t_0, \infty) \times \mathbb{R}^{n+m},
$$

*where $\Lambda(H) \geq 1$ exists for each $0 < H \leq \infty$.*
*When this happens, the equilibrium ($z^*$) of a pulsed CGNN with bidirectional associative memory and a delay of (20) is globally exponentially stable with regard to the function $h$ [17,22,23].*

The Lyapunov function is defined as: [23,24]

$$
V(t, \tilde{z}(t)) = \|\tilde{z}\| = \sum_{i=1}^{n}|\tilde{x}_i(t)| + \sum_{j=1}^{m}|\tilde{y}_j(t)|.
$$

Let $\mathcal{M} = \mathcal{M}(\psi_0) = \Lambda(H)\sup_{-v \leq \zeta \leq 0}\|h(t_0^+, \psi_0(\zeta) - z^*)\|$. Then,

$$
\|h(t, z(t, t_0, \psi_0) - z^*)\| \leq \mathcal{M}\exp(-\mu(t - t_0)), \quad t \geq t_0,
$$

where it is known that $\mathcal{M} \geq 0$ and $\mathcal{M} = 0$ only for $h(t_0^+, \psi_0(\zeta) - z^*) = 0, \zeta \in [-v, 0]$. The last estimate concludes the global exponential stability of the equilibrium state ($z^*$) of (20) with respect to the function ($h$) from [17].

Associative reproduction and dynamic information generation make up the neural network technology for processing unstructured data from various modalities that are being presented. It is based on neural elements in a steady state. Such associative memory data consist of a collection of pieces resembling neurons that are connected in parallel, share an input and an output, and differ from one another in the order of the signals of the synaptic connections on a generalized dendrite. The links each weigh one pound.

A sequence along a trajectory in a multidimensional signal space represents the changed information [21,24,25,25].

## 4. Algorithms of a Stability Model in Cohen–Grossberg-Type Neural Networks

The software implementation of the mathematical model was developed in C programming language. We used OpenMPI technology [26] on a cluster of eight machines, each equipped with four Intel® Xeon® [27] processors.

A neural network based on the generalized CGNN model was used for the software implementation. The implemented model has the following form [28]:

$$
\begin{cases}
\dfrac{x_i(t)}{dt} = -a_i(x_i(t))\left[b_i(x_i(t)) - \sum_{j=1}^{2} c_{ji}f_j(y_j(t)) \right. \\
\qquad\qquad \left. - \sum_{j=1}^{2} d_{ji}g_j(y_j(t-\sigma_j(t))) - I_i\right], \quad t \neq \tau_k(x(t),y(t)), \\
\dfrac{y_j(t)}{dt} = -\hat{a}_j(y_j(t))\left[\hat{b}_j(y_j(t)) - \sum_{j=1}^{2} p_{ij}\hat{f}_i(x_i(t)) \right. \\
\qquad\qquad \left. - \sum_{i=1}^{2} q_{ij}\hat{g}_j(x_i(t-\hat{\sigma}_i(t))) - J_j\right], \quad t \neq \tau_k(x(t),y(t)),
\end{cases}
\tag{22}
$$

with impulse disturbances of the following type:

$$
x(t^+) - x(t) = \begin{pmatrix} -0.5 + \frac{1}{3k} & 0 \\ 0 & -0.5 + \frac{1}{3k} \end{pmatrix} x(t), \quad t = \tau_k(x(t),y(t)), \quad k = 1,2,\ldots,
$$
$$
y(t^+) - y(t) = \begin{pmatrix} -0.5 + \frac{1}{4k} & 0 \\ 0 & -0.5 + \frac{1}{4k} \end{pmatrix} y(t), \quad t = \tau_k(x(t),y(t)), \quad k = 1,2,\ldots,
\tag{23}
$$

where $t > 0$,

$$
\begin{aligned}
&x(t) = \begin{pmatrix} x_1(t) \\ x_2(t) \end{pmatrix}, \quad y(t) = \begin{pmatrix} y_1(t) \\ y_2(t) \end{pmatrix}, \quad I_1 = I_2 = J_1 = J_2 = 0.5, \\
&f_j(y_j) = g_j(y_j) = \frac{|y_j + 0.5| - |y_j - 0.5|}{1.5}, \\
&\hat{f}_i(x_i) = \hat{g}_i(x_i) = \frac{|x_i + 0.5| - |x_i - 0.5|}{1.5}, \\
&0 \leq \sigma_j(t) \leq 1, \quad 0 \leq \hat{\sigma}_i(t) \leq 1, \\
&a_i(x_i) = \hat{a}_j(y_j) = 1.5, \\
&b_1(x_i) = 1.5x_i, \quad b_2(x_i) = 2.5x_i, \\
&\hat{b}_1 = \hat{b}_2 = 1.5y_j, \quad i = 1,2, \quad j = 1,2 \\
&\tau_k(x,y) = |x| + |y| + k + 1, \quad k = 1,2,\ldots.
\end{aligned}
\tag{24}
$$

There are fixed values for the arrays $(C, D, P, Q)$ only in the procedural implementation, as presented in (25). In the implementations using a parallel technique, the initial values of the arrays are set with the minimum value passed as a parameter to the program [26,29].

$$C_{2\times2} = \begin{pmatrix} c_{11} & c_{12} \\ c_{21} & c_{22} \end{pmatrix} = \begin{pmatrix} 1 & -0.5 \\ 0.5 & 0.6 \end{pmatrix}$$

$$D_{2\times2} = \begin{pmatrix} d_{11} & d_{12} \\ d_{21} & d_{22} \end{pmatrix} = \begin{pmatrix} 0.4 & -0.3 \\ 0.3 & 0.5 \end{pmatrix}$$

$$P_{2\times2} = \begin{pmatrix} p_{11} & p_{12} \\ p_{21} & p_{22} \end{pmatrix} = \begin{pmatrix} 0.7 & -0.9 \\ 0.9 & 1 \end{pmatrix} \tag{25}$$

$$Q_{2\times2} = \begin{pmatrix} q_{11} & q_{12} \\ q_{21} & q_{22} \end{pmatrix} = \begin{pmatrix} 0.3 & 0.3 \\ -0.2 & 0.5 \end{pmatrix}$$

All assumptions in Theorem 1 are assumed to be satisfied; therefore, hypotheses 1 to 5 are satisfied [18,19,24]. We set the corresponding constants as follows:

$$L_1 = L_2 = 1, \quad M_1 = M_2 = 1, \quad \hat{L}_1 = \hat{L}_2 = 1, \quad \hat{M}_1 = \hat{M}_2 = 1,$$
$$\underline{a}_i = \overline{a}_i = 1, \quad \hat{\underline{a}}_i = \hat{\overline{a}}_i = 1, \quad B_1 = 2, \quad B_2 = 3, \quad \hat{B}_1 = \hat{B}_2 = 2.$$

A stable point is sought such that there exists a positive number ($\mu$) that satisfies Equation (21). For each value of the four arrays ($C, D, P, Q$), a cyclic calculation is performed for the value of $\mu$, as well as a check for the fulfillment of Equation (21).

## 5. Implementation

All input parameters are entered and saved in a structure for faster access. A link is made to records in different files, depending on the obtained result. All calculated values for the searched function are saved in a common file [25,30]. Only obtained values are stored in a separate file, where the system is stable according to the theorem, i.e., the obtained results are for $\mu > 0$. The algorithm used for validation of the stability point is presented in Figure 1.

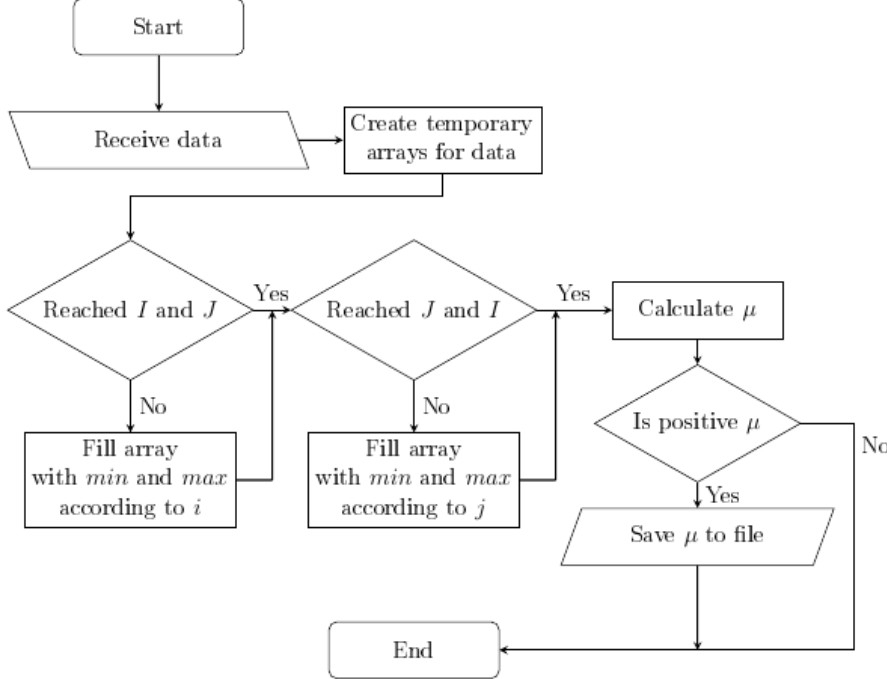

**Figure 1.** Algorithm for validation of the stability point.

In a third separate file, the maximum values of the increasing stability function are recorded, together with the corresponding values for the given arrays. For this purpose, at

the time of calculations, the last calculated highest value of $\mu$ is kept in a structure and compared with the current calculated value. A higher value is recorded and saved as the next reference value to check. Graphically summarized data are presented in Figures 2 and 3.

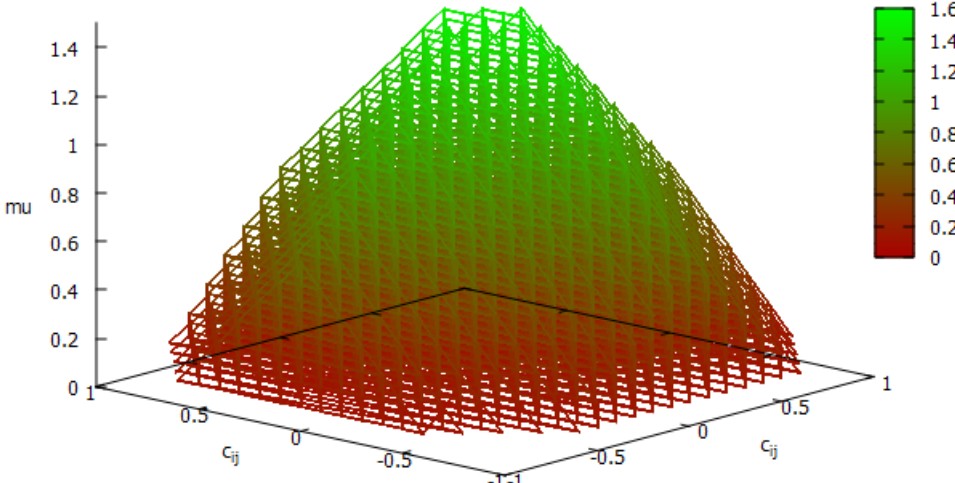

**Figure 2.** Positive $\mu$ values for $C$ .

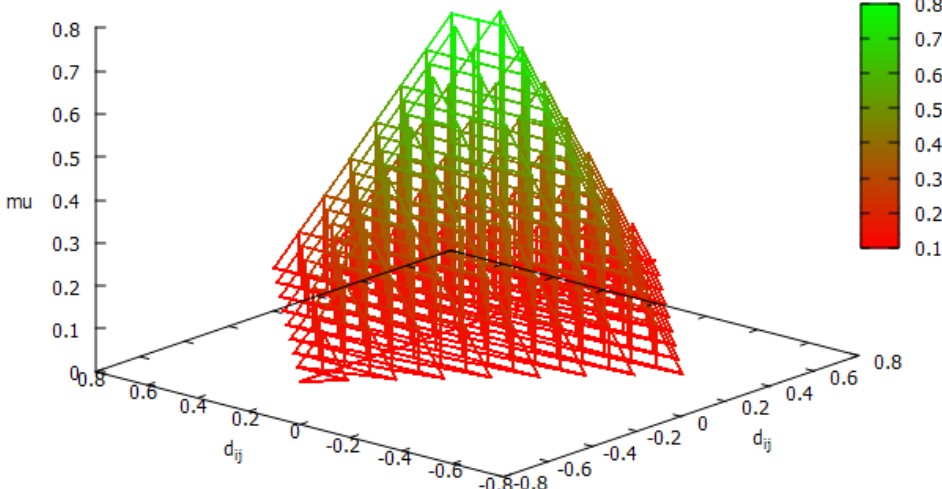

**Figure 3.** Positive $\mu$ values for a $D$ array .

Based on the obtained results, we built an ANN with sixteen input parameters and sixty-four neurons in the hidden layer. Supplied values for the parameters were determined according to Equation (25). The stability results are shown in Figure 4. The blue and red lines represent the response of the first and second layer of the neural network.

The example in Figure 4 explains how impulsive perturbations can be used to influence the stability behavior of CGNNs and demonstrates the usefulness of the theoretical findings that have been proposed. The blue and red colors represent the responses of the first and second layers of the neural network for the period of $8\pi$. After $\pi$ period, we can see that the neural network stabilizes.

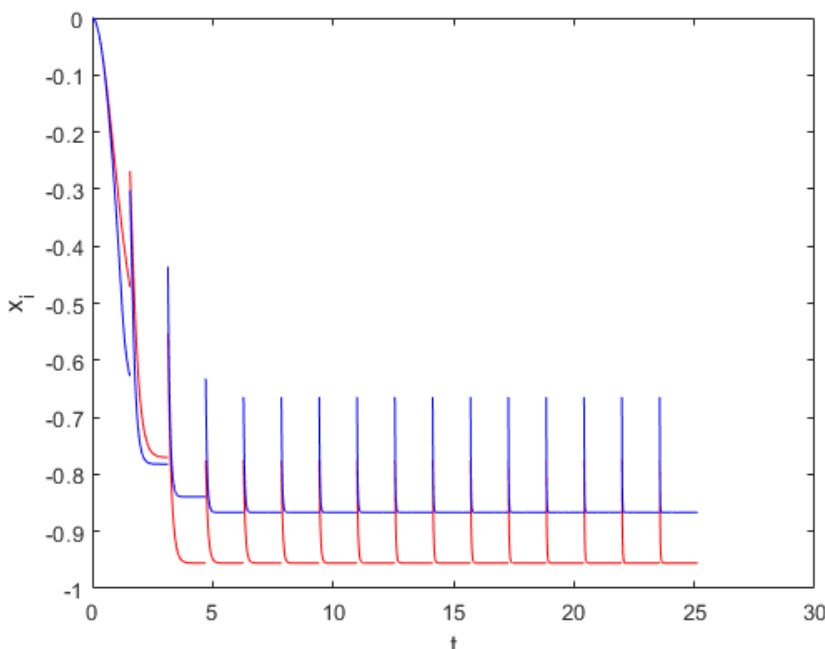

**Figure 4.** Stability result for $x_1$ and $x_2$.

## 6. Conclusions

Based on an analysis of existing intelligent systems, we suggest the use an ANN of the Cohen–Grossberg type to model a sustainable processing of any information in a multidimensional space and a multilevel time structure using dynamic networks with neuro-like elements and a sustainable signal amount. Different modalities using a homogeneous representation of information for a neural network allow for easily integrated information at all levels of the decision-making process. The qualitative properties and global exponential robustness of the solutions with respect to the manifold, as defined by a function for the bidirectional associative memory neural network with time-varying delays, were investigated. A procedural implementation was applied to demonstrate the validity of the obtained criteria for the h stability of the equilibrium state of the model. The software implementation of the mathematical model was developed in C language. The results were examined through a representative sample, since the volume of data was large. Values that meet the requirements of the H-stability theorem, namely positive values of $\mu$, were used, which is a mechanism to account for the statistical properties of the information, along with a nonlinear transformation, and recovery was allowed. Changes in input parameters for the model with all network training were considered to find a stable point. The detailed constraints are discussed in [17] and specified in Section 4. Using the dictionary of elements of the internal structure of the information sequence, an ANN including a dictionary was formed. In future work, we will explore multimodality synchronization by including video and audio data.

**Author Contributions:** Conceptualization, E.G. and I.T.; methodology, E.G. and I.T.; software, I.T.; validation, E.G. and I.T.; formal analysis, E.G. and I.T.; investigation, E.G. and I.T.; resources, E.G. and I.T.; data curation, E.G. and I.T.; writing—original draft preparation, E.G. and I.T.; writing—review and editing, E.G. and I.T.; visualization, E.G. and I.T.; funding acquisition, E.G. All authors have read and agreed to the published version of the manuscript.

**Funding:** This research was funded by the Research and Development Sector of the Technical University of Sofia.

**Institutional Review Board Statement:** Not applicable.

**Informed Consent Statement:** Not applicable.

**Data Availability Statement:** Data sharing not applicable.

**Acknowledgments:** The authors would like to thank the Research and Development Sector of the Technical University of Sofia for the financial support.

**Conflicts of Interest:** The authors declare no conflict of interest. The funders had no role in the design of the study; in the collection, analyses, or interpretation of data; in the writing of the manuscript; or in the decision to publish the results

## Abbreviations

The following abbreviations are used in this manuscript:

| | |
|---|---|
| ANN | Artificial neural network |
| CGNN | Cohen–Grossberg neural network |
| IS | Intelligent system |
| LTM | Long-term memory |
| STM | Short-term memory |
| OpenMPI | Open message-passing interface |

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
