# Peer review of "Information Processing with Stability Point Modeling in Cohen–Grossberg Neural Networks"

_axioms, doi:10.3390/axioms12070612_

Round 1

Reviewer 1 Report

Regarding the paper, the goal of this paper is to investigate efficient ways to represent multi-level structured data using different modalities (images, speech, and text) that can mimic the natural organization of the human brain. To achieve this goal, a variety of theoretical and practical problems need to be solved, including developing a stable mathematical model, an algorithm, and software to process offline data, representing neural networks, and achieving consistent synchronization across various modalities. Therefore, the research proposed by the authors is based on the ANN Cohen-Grossberg model in order to achieve the desired results.

The network's reliability is one of the major difficulties. Instability in the Cohen-Grossberg network can lead to erratic behavior and incorrect results. You have covered this, however, more details and explanation in this regard can be beneficial. Besides, the Cohen-Grossberg network's computational intricacy presents another difficulty. It can be challenging to train and use the network effectively as the computational requirements rise along with the network's size and intricacy. As a result of the Cohen-Grossberg network's sensitivity to the starting conditions, even minor changes in those conditions can result in substantial changes in the network output. Was there limitation in this regard? (Limitations should be addressed in the Conclusion section as well as directions for future research). In my opinion, abbreviations should be provided when appeare in the integral part of the text. Also, the structure of the paper should be given at the end of the Introduction section.

A literature review is a major drawback and it should be significantly improved by taking into consideration recent publications...

Overall, the research is good and the topic is interesting...

Author Response

Dear Sir/Madam,

Thank you very much for your review and the comments, which were very valuable to us for the revision.

Change in input parameters for the model with all network training to find a stable point. The detailed constraints are discussed in 17th paper from bibliography and are specified in section four Algorithms of a stability model in Cohen-Grossberg type neural networks.

We have supplemented the conclusion and given direction for future research.

We have described the structure of the report at the end of the Introduction section.

In order to avoid citation within the citation, citations from the publications of our mentors do not appear in our bibliography, but we have cited them specifically. The Cohen-Grosberg model has been researched since 1983. The history has been extensively reviewed and we have used it to justify the analysis in the publication.

Best regards

Reviewer 2 Report

The authors of the study aim to provide effective ways of representing multi-level structured  information from different modalities (images, speech, text), which allows to natural reproduce the  structure. To achieve the set goals, Cohen-Grossberg type ANN is used with the research methods being based on image processing algorithms, speech, text information, and pattern recognition theory. In terms of methodology, there are multiple layers handled, this is one contribution of the paper.

The application steps of the algorithms and their implementations are provided in an accurate and structured way.

The work is technically sound with interesting subject matter.

My humble recommendations to improve the paper can be outlined as below:

The abstract can be revised to show the main objective of the work in a clearer manner.

The authors can also conduct proofreading, for example, in the title, they can write the prepositions “with” and “in” in lower case as a rule: “Information Processing with Stability Point Modeling in Cohen-Grossberg Neural Networks”.

The contributions can be stated in a more evident way, please.

The authors can show the distinctive part(s) of their work by comparing the study with earlier ones in the literature.

Future directions can also be integrated, if possible.

For your kind information.

Yours faithfully,

Author Response

Dear Sir/Madam,

Thank you very much for your review and the comments, which were very valuable to us for the revision.

We have changed the words "with" and "in" to lower case in the title. 

The abstract was changed and according to us it shows our primary aim.

Comparison with previous works is mentioned in Introduction section.

We have supplemented the conclusion and given direction for future research.

Best regards

Reviewer 3 Report

1. The structure of the article is not described in the introduction.

2. It is not clear what is the novelty of the work. The authors refer to articles by writing out formulas and theorems from them. Where is the main result?

2. Formula (1) does not provide explanations for parameters and functions.

3. In formula (2), not all parameters are explained

4. Not all numbered formulas are referenced

5. Reference to formulas must be in parentheses

6. Formula (7) is not clear, it is necessary to check

7. In formula (7), instead of s1, you need S1

8. There is a gross typo in formula (22), you need to check it.

9. In fig. 4 missing y-axis label

10. It is necessary to explain the graphs.

11. What is the role of each author in writing the article?

12 A careful review of the literature should be carried out. For example, https://ieeexplore.ieee.org/document/1593696 is missing from the bibliography.

The article needs significant revision.

Author Response

Dear Sir/Madam,

Thank you very much for your review and the comments, which were very valuable to us for the revision.

1. We have described the structure of the report at the end of the Introduction section.
2. The development of a robust mathematical model is a major theoretical contribution, given the reliability, sensitivity, and complexity of the CGNN. In training the network and simulating the process, we have found a stable point. In terms of methodology, there are many layers handled; this is one contribution of the paper. We have created a precise and structured algorithm.
3. Parameters in formula (2) are described
4. Parameters in formula (2) are described
5. In our opinion, we do not need to mention the function of all formulas. This is summarized below in the text.
6. The reference to the formulas has been edited
7. & 8. We have revised formula (7)
9. Thank you for formula (22) the mistake is edited
10. Figure 4 is explained
11. The authors have an equal role in writing the article
12. In order to avoid citation within the citation, citations from the publications of our mentors do not appear in our bibliography, but we have cited them specifically. The Cohen-Grosberg model has been researched since 1983. The history has been extensively reviewed and we have used it to justify the analysis in the publication.

The article has been corrected according to the comments of the reviewers.

Best regards

Reviewer 4 Report

This paper has neither theoretical result nor some applications.

The abstract and the conclusion do not correspond to the studied in the paper.

Actually the paper consists of 11 pages. On the first 8 pages some comments on published results as well some known in the literature results are provided. Unfortunately, these results are not clearly presented because many unknown and undefined notations are used. The results of this paper are presented only on pages 8-10. Actually, the authors consider a very special case of the studied in [17] model, just for n=m=2 with a very special impulsive functions. The authors tried to solve this system of equations, to plot the solution on Figure 4, but actually they plotted a scalar function, at the same time the solution of (25) is 4-dimentional vector function.

As an overall, this paper is nto acceptable for publication. Definitely it has to be rejected.

Author Response

Dear Sir/Madam,

The development of a robust mathematical model is a major theoretical contribution, given the reliability, sensitivity, and complexity of the CGNN. In training the network and simulating the process, we have found a stable point. In terms of methodology, there are many layers handled; this is one contribution of the paper. We have created a precise and structured algorithm. The Cohen-Grosberg model has been researched since 1983. The history has been extensively reviewed and we have used it to justify the analysis in the publication.

In Figure 4, we indicate the output of the single neuron ($x_1$) of the network, not the four-dimensional vector function.

Round 2

Reviewer 1 Report

Thank you for inocorporating all the reqs. and improving the manuscript.

Author Response

Thank you for your review.

Reviewer 3 Report

The authors have corrected the comments and therefore the article can be recommended for publication. Check formula references, for example on line 309.

Author Response

Thank you for your review, the formula reference on line 309 was actually a reference to a bibliography item.

Reviewer 4 Report

The authors did not change the manuscript according to any of my comments. The authors wrote"In Figure 4, we indicate the output of the single neuron ($x_1$) of the network, not the four-dimensional vector function." But if they graphed only x_1(t) in Figure 4, I have some complains. The impulses are defined by (23), i.e. x_1(t+)=0.5x_1(t-)+1/(3k). Now look at the graphs at the impulse approximately=3.1. It is visible x_1(3.1-)<0 (it is approximately=-50). Then how x_1(3.1+) is approximately +160?  Definitely, it is not clear.

Definitely, the paper is full with mistakes. 

Author Response

Thanks for the review, we have reviewed all the research results obtained with MatLab. We have revised the article according to all the comments and remarks of the reviewers. We have changed the graphic you commented on. We have demonstrated the stability of neurons in the first layer, namely $x_1$ and $x_2$. We hope to satisfy your comment and apologize for the technical error.

Best regards

Round 3

Reviewer 4 Report

This is my third review to the manuscript. In all my previous reviews I pointed out that the has no theoretical results no practical ones. In the last version, the authors set up e Theore, 1 without any proof. Actually they cited [17,22,23]., which means that there is no theoretical new result in this paper. Also the authors defined on line 245 a function, they called it a Lyapunov function. But this is not a function, it is a functional, the arguments of V are another functions. No domain of these functions are provided, no explanation what are they, etc. How this function is applied? Nothing is given. As a result- theoretical part is absolutely missing or incomplete and incorrect.

What is the difference between (2) and (21)? Why is it necessesarily one and the same system to be written twice.

If the main novelty is the simulation of the model(as the authors insist), then to be considered a network with only neurons is not a real stuff.

The last, the abstract does not correspond to the work in the manuscript:

For example, it is written “the purpose of this article is to identify effective ways of representing multi-level structured information from different modalities (images, speech, text), which allows to natural reproduce the structure, as it happens in the human brain.”Actually, nothing is done.It is written “ the set goal is related to solving a number of problems of a theoretical and practical nature” Again it is not true. First, no theoretical work. Second, no practical work is satisfied. The authors consider only one problem (not a number of problems), only one model with only two neurons and graphed some solutions. That’s it.  The abilities to use CAS or programming language C is not a reason to be accepted this work for publication.

Also, in the abstract it is written” Research methods used in the work are based on image processing algorithms, speech, text information, and pattern recognition theory.” Again it is not true.

As an overall, again I don’t see anything interested and useful for readers. So, there is no reason this paper to be accepted for publication. I definitely advice to be rejected. Also, I would like to suggest authors to send this paper in a journal which scope are algorithms.

Author Response

Dear Mr. Reviewer,
apparently, our corrections to the article did not satisfy you. We do not agree with your recent rude remarks.
1. Since the proof for Theorem 1 is present in the literature we cite, we don't think it's necessary to bother the reader with it. We need it, but it is not the main one, which does not mean that there are no other results in the article.
2. We use the Lyapunov function to prove global exponential stability with respect to the function h.
3. If you are talking about formulas (2) and (21) then we do not see how they are the same system. Formula (2) is the equation of the first layer to normalize the power of the input model, and (21) is an assumption of Theorem 1.
4. In this paper, we identify efficient ways to represent multilevel structured information from different modalities and prove it. "The authors considered only one problem (not a number of problems), only one model with only two neurons, and graphed some solutions." As we have already noted, the authors disagree with the reviewer's notes. We have three positive reviews. Let the journal editors decide whether to publish our article.